# Drug-Related Problems Detected in Complex Chronic Patients by Community Pharmacists of Catalonia: Perception of the Person-Centred Approach Necessity

**DOI:** 10.3390/healthcare12020240

**Published:** 2024-01-18

**Authors:** Berta Torres-Novellas, Pilar Rius, Carlos Figueiredo-Escribá, Eduardo L. Mariño, Pilar Modamio

**Affiliations:** 1Catalan Council of Pharmacists’ Associations (CCFC), 08009 Barcelona, Spain; 2Clinical Pharmacy and Pharmaceutical Care Unit, Department of Pharmacy and Pharmaceutical Technology, and Physical Chemistry, Faculty of Pharmacy and Food Science, University of Barcelona, 08028 Barcelona, Spain

**Keywords:** drug-related problems, community pharmacist, pharmaceutical care, chronic disease, complex chronic patients, patient safety, patient involvement

## Abstract

Community pharmacies are healthcare settings in which pharmacists are in an ideal position to carry out pharmaceutical care. The aim of this study was to analyse the number, type and groups of drugs that caused drug-related problems (DRPs) detected in complex chronic patients who are outpatients, the interventions and actions of community pharmacists and their impact on patient medication adherence. The study was designed as a secondary analysis of a multicentre study in the field of primary healthcare and community pharmacies in Catalonia (Spain). The patients who took part were divided into two groups by the primary care physician depending on whether or not they were considered likely to receive their medication through a monitored dosage system (MDS) based on pre-established criteria. Patients underwent 12 months of follow-up by community pharmacists. The prevalence of DRPs among the studied complex chronic patients was high (n = 689). The most identified DRP was nonadherence (31.20%). In the MDS group, results showed a statistically significant increase of 21% in the number of adherent patients with respect to the baseline visit (*p*-value = 0.0008). Community pharmacists can have an important role in addressing DRPs and optimizing the safety and effectiveness of medications for these patients and in involving them in their own health conditions.

## 1. Introduction

Elderly patients with associated multimorbidity, defined as two or more generally chronic health problems [1], are usually prescribed five or more drugs [2], a fact that increases the risk of inappropriate prescription and drug-related problems (DRPs), including adverse drug reactions and medication errors [3]. Likewise, multiple morbidities increase the probability of hospital admission and readmission, length of stay, and, therefore, the costs of medical care while also reducing quality of life and increasing dependency, polypharmacy and mortality [4]. The fundamental treatment of complex chronic diseases crucially depends on long-term follow-ups with regular reviews of patients [5] since failures due to lack of effectiveness and/or safety in the use of medications have a high cost in terms of health and, consequently, in terms of the public health system [3,6,7]. For this reason, the approach to these complex chronic patients, particularly in terms of the negative impact of DRPs, which is expected to increase in the coming years [4], is a major challenge for the existing models of healthcare provision, which is why it is necessary to guarantee integrated care [8,9,10].

According to the WHO [11], the rational use of medicines is defined as follows: ‘patients receive medications appropriate to their clinical needs, in doses that meet their own individual requirements, for an adequate period of time, and at the lowest cost to them and their community’. In 2022, spending on medicines by the Public Health System of Catalonia (the second most populous of the 17 autonomous communities in Spain) was EUR 1,723,817,443, representing an interannual variation rate of 5.4% compared to the previous year [12]. This high level of spending, which is expected to increase in the coming years (in January 2023, 19.35% of the population of Catalonia was 65 years old or older) [13], requires strategies that guarantee the appropriate use of medicines.

Community pharmacies are healthcare settings in which pharmacists dispense medication to chronic outpatients monthly, which is why they are in an ideal position to carry out pharmaceutical care practice. Pharmaceutical care, according to the Pharmaceutical Care Network Europe, is defined as the pharmacist’s contribution to people’s care to optimise the use of medicines and, thus, improve health outcomes [14]. The World Pharmacy Council, in their report [15], showed the similarities across member countries, such as Australia, Canada, Denmark, Germany, Ireland, New Zealand, Portugal, Spain, the UK and the USA, and highlighted that dispensing, along with the advice-led provision of over-the-counter medicines, are the core activities of the community pharmacy. Therefore, the community pharmacist (CP), in the act of dispensing, should actively intervene in the review and follow-up of the medication of these patients with the aim of preventing, detecting, reporting and resolving DRPs, as well as supporting patients in making appropriate lifestyle decisions [16,17]. Furthermore, this decision-making is in line with the current consensus of person-centred care practice [18], which has potentially significant benefits in terms of healthcare, such as improving health and clinical outcomes, increasing satisfaction with healthcare, improving the efficiency of services and reducing overall costs [19].

Pharmacotherapeutic follow-up with monitored dosage systems (MDS), as a medication repackaging system used for organizing solid dosage forms prescribed to an individual for periods of up to a week, is a professional service provided by community pharmacists once professional judgment and decisions have considered them to be the best option to support patient adherence while providing the information necessary to realise this correctly [20]. In Catalonia, this service is provided following the procedures for action and quality control as described in the Guidance for Pharmacists on the pharmacotherapeutic follow-up use of MDS [21].

Based on everything explained above, this study was designed [22] with the main aim of analysing the number and type of DRPs detected in complex chronic patients on an outpatient basis, the groups of drugs that caused these DRPs and documenting the interventions and actions performed by CPs, particularly in regards to their impact on patient medication adherence with a vision of patient-centred care.

## 2. Materials and Methods

### 2.1. Study Design and Setting

A secondary analysis of a multicentre study was carried out in the field of primary healthcare and community pharmacies in Catalonia (Spain). The study protocol and methodology have already been fully described [22]. Briefly, participating patients received a pharmacy-led intervention for 12 months and, if required, according to benefits based on the pre-determined criteria [21], the medication was organised into monthly MDS. Pharmacists habitually use MDS, also known as multicompartment medication compliance aids (MCAs), as a tool to improve medication adherence in adult chronic patients [23,24]. The criteria for receiving MDS were established by the Department of Health and the Catalan Council of Pharmacists’ Associations’ Guidance for Pharmacists [21] on the pharmacotherapeutic follow-up use of MDS: polypharmacy patients undergoing treatment for chronic disease and with a low degree of adherence, with difficulties in completing treatment, with physical or mental impairments that prevent them from handling medication properly, with alternating, decreasing or irregular or complex dosages, which due to their particular characteristics require adequate follow-up of the medication or patients referred by hospitals or other services which also require specific follow-up. CPs recorded on an online platform all the DRPs they found associated with the medication, as well as all the interventions and referrals to primary care centres.

### 2.2. Study Variables

In terms of patient characteristics, demographic information, medical history, number of medications and devices used were analysed.

Regarding medication-related aspects, the number of DRPs, the types of DRPs according to the Spanish Forum of Pharmaceutical Care in Community Pharmacy (Foro-AF-FC) classification [16] and drugs involved according to the WHO Anatomical Therapeutic Chemical (ATC) classification system were assessed. In the ATC system, the drugs are divided into different groups with five different levels according to the organ or system on which they act and their pharmacological, therapeutic and chemical properties. In the ATC 1st level, the system has 14 main anatomical/pharmacological groups [25].

In addition, medication adherence, measured by the Morisky Green adherence scale [26] and the billing/prescription ratio (Medication Possession Ratio), was also analysed.

Regarding the intervention of the CP, information needs, such as which measures were taken to solve DRPs and whether they were considered solved by the CP or they needed referrals to the primary care centres, were assessed.

### 2.3. Statistical Analysis

A descriptive statistical analysis assuming normality of distribution was performed. Quantitative variables were expressed as mean, standard deviation (SD), maximum, lowest values and percentages, while qualitative variables were expressed as numbers and percentages. For comparisons of qualitative variables, the Chi-squared test was calculated, and for paired samples, McNemar’s test was evaluated. For quantitative variables for two subgroups, the Student-*t* test was used.

Statistical significance was expressed as *p* < 0.05. IBM^®^ SPSS^®^ Statistics 29.0 (IBM Corp., Armonk, NY, USA) software was used to evaluate the data.

### 2.4. Ethics Approval and Informed Consent

The study protocol was approved by the IDIAP Jordi Gol Clinical Research Ethics Committee (IRB00005101) and by the University of Barcelona Bioethics Committee (IRB00003099). Informed consent documentation was obtained from patients both through the primary care team and the CP. The patients only signed if they agreed to be included in the study and to everything else that it involved.

## 3. Results

### 3.1. Patient Profile

Table 1 shows the patients’ profiles evaluated in their baseline state. For the assessment of DRPs, a total of 146 patients were collected and divided into two groups, depending on whether they were considered likely to receive the medication organised as MDS (n = 95) according to benefits based on the previously described criteria or not (n = 51).

### 3.2. Prevalence and Type of Drug-Related Problem

Regarding the type and number of DRPs detected during the study period, a total of 689 DRPs were identified; 540 (mean = 5.68, SD = 15.32) were reported in the MDS group and 149 (mean = 2.92, SD = 5.11) in the no MDS group (*p* = 0.1144). The DRP classification for each group is shown in Table 2.

Also, 412 DRPs were detected in women and 276 in men, which represents 1.45 times more DRPs detected in women. If we look at the number of DRPs depending on age group, patients aged between 71 and 80 years old had 1.41 times more DRPs compared to those aged 81–90 years.

All registered DRPs were associated with the WHO ATC classification system of the drugs. Drugs for the nervous system (N) were those that caused the most DRPs (n = 246; 35.70%), followed by those for the cardiovascular system (C) (n = 134; 19.45%), digestive tract and metabolism (A) (n = 107; 15.53%) and respiratory system (R) (n = 83; 12.05%). Figure 1 shows all the DRPs detected that were associated with the first level ATC group of the drug.

### 3.3. Adherence

Table 3 shows results related to the assessment of medication adherence in both groups during the study period. In the MDS group, the number (%) of adherent patients according to the Morisky Green test was 49 (51.6%) at the baseline visit and 69 (72.6%) at the final visit, a statistically significant increase of 21% in the number of adherent patients with respect to the baseline visit (*p*-value = 0.0008). With regard to the Medication Possession Ratio for this same group, categorised according to the degree of medication adherence, a growing tendency to greater adherence could be observed at the final visit compared to the baseline visit, while at the baseline visit, the highest concentration of patients was in the ≥80% and <60% strata, with 75.8% and 13.7%, respectively, at the final visit, the majority of patients managed to move towards the (70%, 80%) and ≥80% strata, with 17.9% and 64.2% of the patients, observing a decrease in the stratum of <60% with 10.5%. In the no MDS group, the number (%) of compliant patients according to the Morisky Green test at the baseline visit and at the final visit were 31 (60.8%) and 39 (76.5%), respectively. Despite the increase of 15.7% observed in this group, this difference was not statistically significant (Table 2). Consistent with the Morisky Green scale, in the Medication Possession Ratio categorised according to the degree of adherence, an increase was observed in the categories of greater medication adherence at the final visit compared to the baseline visit; specifically, the patients with 70–80% of possession were 17.6% at the final visit and 11.8% at the baseline visit, and patients ≥ 80% increased from 62.7% at the baseline visit to 66.7% at the final visit.

### 3.4. Pharmacists’ Interventions

Regarding the CP interventions performed on the patients, a total of 728 interventions were recorded: 572 on patients in the MDS group and 156 on patients in the no MDS group. The results are provided in Table 4.

A total of 63 referrals to primary care and communication with a physician of the DRPs were registered. Regarding the referral actions, it could be observed that in the MDS group, there was a higher percentage of treatment change proposals compared to the no MDS group (MDS: 28 (73.7%), no MDS: 12 (48.0%)).

Table 5 compares the mean of the number of DRPs, number of recorded interventions and number of DRPs that caused referral between adherent and non-adherent patients according to the Morisky Green scale at a first visit to the community pharmacy.

## 4. Discussion

This is the first published study that analyses data about complex chronic patients in the community pharmacy setting in Catalonia (Spain), which has 7,899,056 inhabitants [27], an aging population and a low birth rate. Regarding the baseline characteristics of the patients included in the study, no statistically significant differences were found in the distribution of the sex variable and the number of medications between the two groups. However, statistically significant differences were found between the mean age in the MDS and no MDS groups (*p*-value = 0.0075). This could be attributed to the fact that since the distribution between the two groups was made based on the decision of the primary care physician, older patients often met more of the criteria to receive their medication using an MDS since it is intended to facilitate patient medication adherence [19]. In our study, 74.66% of the patients were 75 years of age or older, all were prescribed four or more medications, and the sex ratio was balanced, a patient sample comparable to a German study [28] in which DRPs were identified by CPs.

In our study, CPs detected a considerable number of DRPs (n = 689) in 146 patients, which represents an average of 4.72 DRPs per patient. These findings are similar to other published studies [29,30]. Although no statistically significant differences were found regarding the number of DRPs detected per patient between groups (MDS and no MDS), our results show a higher number of DRPs in the MDS group. This fact could also be related to the age of patients in each group, given that those in the MDS group were older and age has been associated with an increase in medication-related problems due to age-related physiological changes and multiple medications taken for multiple comorbidities [31,32].

The largest proportion of DRPs identified was nonadherence (31.20%), followed by drug interactions and inappropriate administration (11.90%), similar to findings from other studies [28,30,33]. To solve DRPs, 728 interventions were undertaken and registered, and 8.65% of these involved referrals to a primary care physician, but, as other studies have shown [29,34], most DRPs were solved by pharmacists directly. This fact is especially relevant because it highlights the importance of pharmacotherapeutic follow-up by the CPs and the importance of good communication between CPs and primary care physicians.

Although some published articles do not always consider MDSs to be the most appropriate solution to improving patient medication adherence [35,36,37], in our study, the number of adherent patients in the MDS group increased significantly between the first and last visit (*p*-value: 0.0008) after 12 months of follow-up, which shows that MDSs could be a useful tool alongside the pharmaceutical care provided for elderly outpatients with chronic diseases. A similar trend was reported in a study conducted by Martin et al. [38], in which medication adherence improved among the included patients after preparing the MDSs and with the added value of a pharmaceutical follow-up. However, other published studies show an increase in medication adherence among complex chronic patients after a structured pharmaceutical intervention without preparing MDSs [39,40,41]. In any case, these findings are important because in all of these studies, the CP carried out a protocol-based and tailored follow-up with patients, involving and empowering them in the control of their disease through educational programmes delivered by pharmacists, and this also leads to an increase in patient medication adherence [38,39,40,41]. In addition, pharmacists are able to identify relevant clinical information that allows them to detect possible DRPs, and this could protect patients from negative outcomes associated with medication in terms of needs, effectiveness and safety [42].

Our study also found that the drugs most commonly involved in problems were those to treat the nervous system, the cardiovascular system, the alimentary tract and metabolism, in line with a German study [43], which also examined medication-related problems identified by CPs.

In addition, statistically significant differences were found between the number of DRPs (*p* = 0.014) and the number of pharmaceutical interventions (*p* = 0.018) between the adherent and non-adherent patient groups. In both cases, these were higher in non-adherent patients. This fact could indicate that in complex chronic patients, a possible strategy to improve adherence could focus on the person-centred care approach [44] and, thus, avoid future DRPs and derive pharmaceutical interventions that could be considered an effective strategy to save direct costs in the healthcare system, according to some published studies [45,46].

Finally, our study further demonstrates the importance of the clinical role of CPs in preventing potential negative health outcomes, such as preventing medication errors and ensuring medication safety, particularly in complex chronic patients.

One of the limitations of the study was the lack of information regarding referred patients because the channel of communication, established through electronic prescriptions, between community pharmacies and primary care centres was not effective. Greater communication would have allowed the establishment of a continuity of care for complex chronic patients. Moreover, due to the limited sample size (especially in the no MDS group), the results obtained in this study cannot be guaranteed to be conclusive. Furthermore, additional research is required on pharmacy-based medication reviews and pharmacy practices towards a person-centred care approach besides the efficacy of MDSs for chronic complex patients. Therefore, it is recommended that these limitations be taken into account when interpreting the results of this study.

## 5. Conclusions

This study shows that CPs identified a significant percentage of DRPs in elderly outpatients with chronic diseases that might otherwise have remained undetected. MDSs could be a useful tool alongside pharmaceutical care and the pharmacist’s role in resolving DRPs, such as patient medication adherence. Most of the DRPs detected, especially in non-adherent patients, were solved by the CPs providing tailored information about medications and offering health education to patients, thus involving them in their own health conditions.

## Figures and Tables

**Figure 1 healthcare-12-00240-f001:**
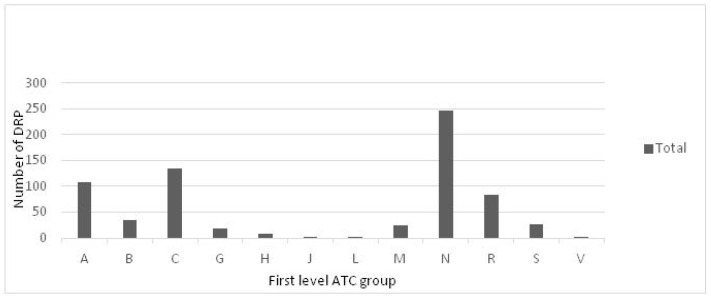
Drug-related problems (DRPs) associated with the drug’s 1st level ATC group [25]. The ATC system has fourteen main anatomical/pharmacological groups (1st level). A: alimentary tract and metabolism; B: blood and blood-forming organs; C: cardiovascular system; D: dermatologicals; G: genitourinary system and sex hormones; H: systemic hormonal preparations, excluding sex hormones and insulins; J: antiinfective for systemic use; L: antineoplastic and immunomodulating agents; M: musculo-skeletal system; N: nervous system; P: antiparasitic products, insecticides and repellents; R: respiratory system; S: sensory organs; V: various.

**Table 1 healthcare-12-00240-t001:** Patients’ profiles distributed into two groups: monitored dosage systems (MDS) or no MDS.

	MDS (n = 95)	No MDS (n = 51)	*p*-Value
Age, mean (±SD)	80 (7.4)	77 (6.6)	0.0075 *
Age (stratified), n (%)			
51–64	3 (3.20)	3 (5.90)	0.0205 *
65–74	18 (18.90)	13 (25.50)	
75–84	40 (42.10)	29 (56.90)	
≥85	34 (35.80)	6 (11.80)	
Men, n (%)	44 (46.30)	28 (54.90)	0.4147
No. of medications, mean (±SD)	11.6 (4.1)	10.7 (4.7)	0.1067
No. of medications (stratified), n (%)			
<5 medications	5 (5.30)	6 (11.80)	0.5196
5–10 medications	46 (48.40)	25 (49.00)	
11–14 medications	25 (26.30)	11 (21.60)	
>14 medications	19 (20.00)	9 (17.60)	
Patients with hypertension, n (%)	77 (81.10)	47 (92.20)	0.1222
Patients with metabolic disorders (Lipoproteins and lipidaemia), n (%)	58 (61.10)	34 (66.70)	0.6241
Patients with diabetes mellitus, n (%)	51 (53.70)	27 (52.90)	1
Patients with osteoarthritis, n (%)	42 (44.20)	21 (41.20)	0.859
Patients with heart failure, n (%)	29 (30.50)	11 (21.60)	0.3359
Patients with extremity or joint pain, n (%)	22 (23.20)	8 (15.70)	0.3951
Patients with severe depressive disorders, n (%)	24 (25.30)	8 (15.70)	0.2611
Patients with COPD, n (%)	14 (14.70)	12 (23.50)	0.2726
Patients with osteoporosis, n (%)	17 (17.90)	5 (9.80)	0.289
Patients with anaemia, n (%)	19 (20.00)	6 (11.80)	0.3035
Patients with rheumatic pathology, n (%)	15 (15.80)	6 (11.80)	0.6793

* Differences between groups are statistically significant, *p*-value < 0.05; Statistical tests used: Student’s *t*-test and Chi-Square test. COPD: chronic obstructive pulmonary disease.

**Table 2 healthcare-12-00240-t002:** Drug-related problems (DRPs) detected by community pharmacists as totals and distributed into two groups, monitored dosage systems (MDS) or no MDS, according to the Foro AF-FC classification [16].

Type of DRP	MDS	No MDS	Total
1: Inappropriate drug administration, n (%)	59 (10.9)	23 (15.4)	82 (11.90)
2: Personal characteristics, n (%)	26 (4.8)	5 (3.4)	31 (4.50)
3: Inappropriate drug conservation, n (%)	52 (9.6)	0 (0.0)	52 (7.55)
4: Contraindication, n (%)	6 (1.1)	0 (0.0)	6 (0.87)
5: Inadequate dose, schedule and/or duration, n (%)	50 (9.3)	14 (9.4)	64 (9.29)
6: Duplications, n (%)	8 (1.5)	7 (4.7)	15 (2.18)
7: Dispensing errors, n (%)	3 (0.6)	0 (0.0)	3 (0.44)
8: Errors in prescription, n (%)	4 (0.7)	0 (0.0)	4 (0.58)
9: Non-adherence, n (%)	182 (33.7)	33 (22.1)	215 (31.20)
10: Drug interactions, n (%)	57 (10.6)	25 (16.8)	82 (11.90)
11: Unnecessary drug therapy, n (%)	19 (3.5)	5 (3.4)	24 (3.48)
12: Other health problems affecting medication, n (%)	8 (1.5)	7 (4.7)	15 (2.18)
13: Probability of adverse drug reactions, n (%)	40 (7.4)	16 (10.7)	56 (8.13)
14: Undertreated health problem, n (%)	6 (1.1)	6 (4.0)	12 (1.74)
15: Others, n (%)	20 (3.7)	8 (5.4)	28 (4.06)

**Table 3 healthcare-12-00240-t003:** Assessment of medication adherence in the two groups: monitored dosage systems (MDS) or no MDS.

	MDS (n = 95)			No MDS (n = 51)		
	Baseline Visit	Last Visit	*p*	Baseline Visit	Last Visit	*p*-Value
Morisky Green test—medication adherent patient, n (%)	49 (51.6)	69 (72.6)	0.0008 *	31 (60.8)	39 (76.5)	0.0990
Possession index—% medication adherence, (<60%), n (%)	13 (13.7)	10 (10.5)	0.1242	5 (9.8)	6 (11.8)	0.4131
Possession index—% medication adherence, (60%, 70%), n (%)	4 (4.2)	7 (7.4)		8 (15.7)	2 (3.9)	
Possession index—% medication adherence, (70%, 80%), n (%)	6 (6.3)	17 (17.9)		6 (11.8)	9 (17.6)	
Possession index—% medication adherence, (≥80%), n (%)	72 (75.8)	61 (64.2)		32 (62.7)	34 (66.7)	
Possession index—% medication adherence (continuous) **, mean (±SD)	84.7 (20.4)	83.1 (18.5)	0.4052	82.4 (19.4)	83.7 (19.7)	0.6754

* Differences between groups are statistically significant, *p*-value < 0.05. Statistical tests used: Student’s *t*-test and McNemar’s test. ** Mean value and SD of the possession index at the base and final moment of the group.

**Table 4 healthcare-12-00240-t004:** Community pharmacists’ interventions in the two groups: monitored dosage systems (MDS) or no MDS.

Type of Intervention	MDS (n = 572)	No MDS (n = 156)
Facilitate PDI, n (%)	261 (45.6)	78 (50.0)
Offer health education, n (%)	215 (37.6)	45 (28.8)
Propose lifestyle changes, n (%)	57 (10.0)	8 (5.1)
Report in pharmacovigilance, n (%)	1 (0.2)	0 (0.0)
Refer to the primary care physician and communicate DRPs, n (%)	38 (6.6)	25 (16.0)

DRP: Drug-related problem; PDI: Personalised Drug Information.

**Table 5 healthcare-12-00240-t005:** Comparison of the number of drug-related problems (DRPs), interventions and DRPs that caused referral between adherent and non-adherent patients.

	Non-Adherent (n = 66)	Adherent (n = 80)	*p*-Value
No. of DRPs, mean (±SD)	7.83 (17.47)	2.15 (6.04)	0.014 *
No. of interventions, mean (±SD)	8.08 (18.12)	2.44 (6.05)	0.018 *
No. of DRPs that caused referral, mean (±SD)	0.56 (1.15)	0.33 (0.671)	0.145

* Differences between groups are statistically significant, *p*-value < 0.05. Statistical test used: Student’s *t*-test.

## Data Availability

The data that support the findings of this study are available upon request from the corresponding author. The data are not publicly available due to privacy or ethical restrictions.

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
