# Peer review of "Drug-Related Problems Detected in Complex Chronic Patients by Community Pharmacists of Catalonia: Perception of the Person-Centred Approach Necessity"

_healthcare, 2024, doi:10.3390/healthcare12020240_

Round 1
Reviewer 1 Report
Comments and Suggestions for Authors
1. It is essential for the authors to provide an introduction to the Monitored Dosage System (MDS) utilized in the study. This background information will help readers understand the context and relevance of the MDS to the research.
2. The abbreviation "CP" mentioned in the abstract needs to be defined upon its first occurrence. This clarification is crucial for readers unfamiliar with the term.
3. Tables 1, 2, and 3 require reformatting. The repetitive appearance of "n" and "%" within the tables currently leads to confusion. A clearer presentation of this data would significantly enhance readability and comprehension.
4. For each table, the specific statistical test methods employed in the study should be explicitly stated. This detail is vital for readers to understand the analytical approach and assess the validity of the results.
5. For Figure 1:
- A legend explaining the meaning of each alphabetic abbreviation in Figure 1 is necessary.
- The figure should include a title for overall context.
- Labels for the x and y-axes need to be added to aid in data interpretation.
Author Response
- It is essential for the authors to provide an introduction to the Monitored Dosage System (MDS) utilized in the study. This background information will help readers understand the context and relevance of the MDS to the research.
Response: Many thanks to the Reviewer’s commentary that it will improve the introduction. So, we have added a paragraph in the introduction explaining the MDS service provided by the community pharmacist (in line 64).
- The abbreviation "CP" mentioned in the abstract needs to be defined upon its first occurrence. This clarification is crucial for readers unfamiliar with the term.
Response: Yes, thank you very much for highlighting this point. We wrote all the word in the abstract (community pharmacists) as its first occurrence.
- Tables 1, 2, and 3 require reformatting. The repetitive appearance of "n" and "%" within the tables currently leads to confusion. A clearer presentation of this data would significantly enhance readability and comprehension.
Response: Thank you so much for your commentary. We have improved, as you suggested, the three first tables of the article.
- For each table, the specific statistical test methods employed in the study should be explicitly stated. This detail is vital for readers to understand the analytical approach and assess the validity of the results.
Response: Thank you so much for your commentary. We have added, according to your recommendation, the specific statistical methods employed in tables 1, 3 and 5 (tables 2 and 4 are descriptive).
- For Figure 1:
- A legend explaining the meaning of each alphabetic abbreviation in Figure 1 is necessary.
- The figure should include a title for overall context.
- Labels for the x and y-axes need to be added to aid in data interpretation.
Response: Thank you for all these suggestions that we have already considered in the new version of the article. So, we have added a legend explaining the meaning of each alphabetic abbreviation of the Anatomical Therapeutic Chemical (ATC) classification system (1st level) and the labels for the “x” and “y” axes. The figure already includes a title for overall context: ‘Figure 1. Drug related problems (DRPs) associated with the drug’s 1st level ATC group.’, although we have moved it to the beginning of the figure.
Reviewer 2 Report
Comments and Suggestions for Authors
Thank you for inviting me to review this work entitled "Drug-related problems detected in complex chronic patients by community pharmacists of Catalonia: Perception of the person- centred approach necessity" which aimed to analyze the number, type and groups of drugs that caused drug related problems (DRPs) detected in complex chronic patients who are outpatients, the interventions and actions of community pharmacists and their impact on patient medication adherence.
Actually, it is a very interesting study and I believe this concept of drug related problem handling by community pharmacists not well-established for healthcare professionals and deserves to be highlighted.
Some comments were raised and need to be addressed:
General comment
· English language proofreading is required.
Abstract:
· Please follow the authors guidelines regarding the abstract, it should be structured.
· Please clarify the recruitment method in the abstract.
· Please add one sentence at the end of the abstract to highlight the clinical Implications of the study.
· Please add to the introduction the context of Catalonia of Spain in terms of DRP and pharmacists’ roles/structure. Show why we need to publish this work in an international journal.
· Please add more details about “WHO Anatomical Therapeutic Chem- 91 ical (ATC) classification system” in the method section to pave the path for the reader to understand the tool.
· Elaborate more in the statistical analysis section
· Figure 1 is not clear to the reader and is misleading.
· Please add the used statical test as a footnote for each table
· I believe the results section could be rearranged to answer the research questions and the flow of the methods.
· In discussion, line 240, what do you mean by? “This fact could indicate that in complex chronic patients the focus should be placed on involving them, as actives partners, in the control of their disease” ?
· Please add a paragraph at the end of the discussion before the limitations about clinical implications
Comments on the Quality of English Language
Minor English editing is required
Author Response
- English language proofreading is required.
Response: The document has been revised by a native English-speaking professional translator/corrector from the Language Advisory & Translation Unit of the Language Service at the Universitat Autònoma de Barcelona. Please find attached the CERTIFICATE OF LANGUAGE REVISION FOR THE SUBMITTED ARTICLE. If still, you recommend that it must be reviewed again, we will have to send again the document to the translation center of the Autonomous University of Barcelona, but it will take a little more time (normally between 2 and three weeks).
Abstract:
- Please follow the authors guidelines regarding the abstract, it should be structured.
Response: many thanks to the comment. The abstract should be a total of 200 word maximum. We try to eliminate the least relevant information and add the points recommended to us in the following suggestions. The abstract follows the style of structured abstracts: background, methods, results and conclusion as the authors guidelines recommend.
- Please clarify the recruitment method in the abstract.
Response: many thanks to the Reviewer. We introduced some information to clarify the methods section in the abstract.
- Please add one sentence at the end of the abstract to highlight the clinical Implications of the study.
Response: many thanks for the recommendation. We have added this sentence at the end of the abstract: ‘Community pharmacists can have an important role in addressing the problems and optimizing the safety and effectiveness of therapies for these patients, involving them in their own health conditions.’ Besides, we have restructured the abstract to follow the standards of the authors' guidelines.
- Please add to the introduction the context of Catalonia of Spain in terms of DRP and pharmacists’ roles/structure. Show why we need to publish this work in an international journal.
Response: many thanks to the Reviewer for the important appreciation. We mentioned in the introduction the World Pharmacy Council where Spain is one of their countries along with Australia, Canada, US, UK, Portugal, etc. So, we have cited the 2023 report from the World Pharmacy Council which shows that in the global healthcare landscape, community pharmacies are at the forefront, facing both challenges and opportunities that often transcend borders making them globally relevant (line 63).
- Please add more details about “WHO Anatomical Therapeutic Chemical (ATC) classification system” in the method section to pave the path for the reader to understand the tool.
Response: many thanks to the Reviewer for this suggestion. We have added more details about ATC system in the methods section (line 100).
- Elaborate more in the statistical analysis section
Response: Thank you so much for your commentary. We have rewritten this section for better understanding.
- Figure 1 is not clear to the reader and is misleading.
Response: thank you for your suggestions to improve Figure 1. We have added a legend explaining the meaning of each alphabetic abbreviation of the Anatomical Therapeutic Chemical (ATC) classification system (1st level) and the labels for the “x” and “y” axes to improve the understanding to the reader.
- Please add the used statistical test as a footnote for each table
Response: Thank you so much for your commentary. We have added, as you suggested, the specific statistical methods employed in tables 1, 3 and 5 (tables 2 and 4 are descriptive).
- I believe the results section could be rearranged to answer the research questions and the flow of the methods.
Response: Thank you so much for your suggestion. The results section is divided in: 1) Patient’s profile, 2) Prevalence and type of DRP, 3) Adherence, 4) Pharmacists intervention. Finally, we reorganized the methods section because we consider that the results should be shown in this order.
- In discussion, line 240, what do you mean by? “This fact could indicate that in complex chronic patients the focus should be placed on involving them, as actives partners, in the control of their disease”?
Response: Thank you for your appreciation. The answer is yes. As in non-adherent patients we observed a greater number of DRPs and a greater number of pharmaceutical interventions, the focus should be placed on increasing adherence in these types of patients. To increase the adherence one possible strategy is to recognizes the patient’s experiences and involve them as an active partner in the control of their care (person-centred approached necessity). We have modified the sentence (line 295).
- Please add a paragraph at the end of the discussion before the limitations about clinical implications
Response: Thank you so much for your commentary. We have added, as you suggested, a paragraph resuming clinical implications about solving and detecting DRP.

Round 2
Reviewer 1 Report
Comments and Suggestions for Authors
The authors have carefully addressed my concerns.
Reviewer 2 Report
Comments and Suggestions for Authors
Thank you for addressing the comments.